# NRAS Mutations May Be Involved in the Pathogenesis of Cutaneous Rosai Dorfman Disease: A Pilot Study

**DOI:** 10.3390/biology10050396

**Published:** 2021-05-02

**Authors:** Kuan-Jou Wu, Shu-Hao Li, Jia-Bin Liao, Chien-Chun Chiou, Chieh-Shan Wu, Chien-Chin Chen

**Affiliations:** 1Department of Dermatology, Kaohsiung Veterans General Hospital, Kaohsiung 813, Taiwan; 100311043@gms.tcu.edu.tw (K.-J.W.); dermawu@vghks.gov.tw (C.-S.W.); 2Department of Dermatology, Chang Gung Memorial Hospital, Linkou, Taipei 333, Taiwan; 100311005@gms.tcu.edu.tw; 3School of Medicine, National Yang-Ming University, Taipei 112, Taiwan; jbliao@gmail.com; 4Department of Pathology and Laboratory Medicine, Kaohsiung Veterans General Hospital, Kaohsiung 813, Taiwan; 5Department of Dermatology, Ditmanson Medical Foundation Chia-Yi Christian Hospital, Chiayi 600, Taiwan; 07063@cych.org.tw; 6Department of Dermatology, Faculty of Medicine, College of Medicine, Kaohsiung Medical University, Kaohsiung 807, Taiwan; 7Department of Pathology, Ditmanson Medical Foundation Chia-Yi Christian Hospital, Chiayi 600, Taiwan; 8Department of Cosmetic Science, Chia Nan University of Pharmacy and Science, Tainan 717, Taiwan

**Keywords:** skin, Rosai–Dorfman, genetic, KRAS, NRAS, BRAF, skin cancer, biomarker

## Abstract

Background: Purely cutaneous Rosai-Dorfman disease (RDD) is a rare histiocytic proliferative disorder limited to the skin. To date, its pathogenesis remains unclear. Owing to recent findings of specific mutations in the mitogen-activated protein kinase/extracellular signal-regulated kinase (MAPK/ERK) pathway in histiocytic proliferative disorders, it provides a novel perspective on the pathomechanism of cutaneous RDD. We aim to investigate the genomic mutations in MAPK/ERK pathway in cutaneous RDD. Methods: We retrospectively recruited all cases of cutaneous RDD from two hospitals in Taiwan from January 2010 to March 2020 with the clinicopathologic features, immunohistochemistry, and treatment. Mutations of neuroblastoma RAS viral oncogene homolog (*NRAS*)*,* Kirsten rat sarcoma 2 viral oncogene homolog (*KRAS*), and v-raf murine sarcoma viral oncogene homolog B1 (*BRAF*) in MAPK/ERK pathway were investigated by the highly sensitive polymerase chain reaction with Sanger sequencing. Results: Seven patients with cutaneous RDD were recruited with nine biopsy specimens. The median age was 46 years (range: 17–62 years). Four of seven patients (57.1%) received tumor excision, while the other three chose oral and/or topical or intralesional steroids. *NRAS* mutation was detected in 4 of 7 cases (4/7; 51.7%), and *NRAS* A146T was the most common mutant point (*n* = 4/7), followed by *NRAS* G13S (*n* = 2/7). There is no *KRAS* or *BRAF* mutation detected. Conclusions: We report the *NRAS* mutation is common in cutaneous RDD, and *NRAS* A146T was the most frequent mutation in this cohort. Mutations in the *NRAS* gene can activate the RAS/MAPK signaling and have been reported to be associated with various cancers. It indicates that *NRAS* mutation in MAPK/ERK pathway may involve the pathogenesis of cutaneous RDD.

## 1. Introduction

Rosai–Dorfman disease (RDD), also known as “sinus histiocytosis with massive lymphadenopathy”, is a rare histiocytic proliferative disorder. It was first described in 1965 by Destombes, and later recognized as a distinct clinicopathologic entity of pale-staining histiocytes in 1969 by Rosai and Dorfman [1,2]. RDD was traditionally classified as non-Langerhans cell histiocytosis (non-LCH) [3], and now belongs to the “R” group of the 2016 revised histiocytosis classification [4]. There is a predilection for young adults with a relatively male predominance [5,6]. Bilateral painless cervical lymphadenopathy is the most prominent clinical manifestation, frequently associated with fever, leukocytosis, neutrophilia, and elevated erythrocyte sedimentation rate (ESR) [5,7]. Most of RDD commonly involved in lymph nodes, but extranodal involvements including the skin, paranasal sinuses, nasal and subcutaneous tissues, eyes, eyelids, and bone, ordered by the prevalence, were not uncommon and reported to occur in approximately 40% of cases with variable symptoms [7,8]. Typical or nodal RDD can have cutaneous manifestation concurrently, but a purely cutaneous manifestation is extremely rare. Some reports had studied purely cutaneous RDD and found it distinct from typical RDD in epidemiology, systemic clinical symptoms, and histopathology [9,10,11,12,13]. Therefore, purely cutaneous RDD can be considered as a distinct entity, and it also comes to our notice that the pathogenesis of purely cutaneous RDD is identical to typical RDD.

The etiology of RDD remains unclear. Some studies suggest that viral infections such as herpesvirus-6 and Epstein-Barr virus can be a precipitating event causing host immune dysregulation [14,15,16], though there are still some conflicts [17,18]. Familial RDD with germline mutations in SLC29A3 was reported, but only in few cases [19]. In the last decade, great breakthroughs of molecular pathology in histiocytic disorders provided a novel aspect of the pathogenesis of these diseases. The specific mutations in the mitogen-activated protein kinase/extracellular signal-regulated kinase (MAPK/ERK) pathway were identified, including v-raf murine sarcoma viral oncogene homolog B1 (*BRAF*) (V600E), Mitogen-Activated Protein Kinase Kinase 1 (*MAP2K1)*, neuroblastoma RAS viral oncogene homolog (*NRAS*), and Kirsten rat sarcoma 2 viral oncogene homolog (*KRAS*), which indicated that the MAPK/ERK pathway may play a key role in the pathomechanism of LCH, Erdheim–Chester disease (ECD), and even other non-LCH diseases [20,21,22,23,24,25,26,27,28]. Some studies, therefore, investigated for gene mutations in RDD. To date, approximately 33–50% of cases were detected with the genomic mutation involving *KRAS, NRAS,* serine/threonine-protein kinase A-Raf *(ARAF),* and *MAP2K1* in RDD [21,28,29,30,31,32,33], and only two cases with *BRAF* mutation were reported [34,35]. However, the results of the studies were different from each other. Furthermore, none of them focused on purely cutaneous RDD.

Due to the variable results of the previous studies and the distinct entity of purely cutaneous RDD, herein, we assessed 7 cases of purely cutaneous RDD for reported genomic mutations in the MAPK/ERK pathway. To our knowledge, this is the first study to focus on the genomic mutations of purely cutaneous RDD.

## 2. Materials and Methods

### 2.1. Patient Enrollment

This case study had retrospectively reviewed all medical records with a pathological diagnosis of cutaneous RDD in the Kaohsiung Veterans General Hospital and the Ditmanson Medical Foundation Chia-Yi Christian Hospital in Taiwan during the period from January 2010 to March 2020. All cases were histopathologically confirmed, and would be included if the following criteria were all met: (1) a pure cutaneous lesion; (2) proliferation of RD cells; (3) the RD cells were immunohistochemically positive for S100 and CD68 but negative for CD1a. Data were retrospectively collected and included age at presentation, sex, clinical symptoms, histopathological findings, immunohistochemical profiles, clinical management, and outcome. Patients with systemic manifestations or lymphovascular involvements were excluded. The study was approved by the Ethics Committees of Ditmanson Medical Foundation Chia-Yi Christian Hospital (CYCH-IRB-2021008, on 20 January 2021) and was conducted under the ethical guidelines of the Declaration of Helsinki.

### 2.2. Histopathology and Immunohistochemistry

The specimens of resected primary tumors were retrieved and examined with hematoxylin and eosin staining and immunohistochemistry including at least S-100, CD68, CD1a, and LCA to confirm the histopathologic diagnosis. With paraffin-embedded blocks of all RDD patients, formalin-fixed, paraffin-embedded sections were prepared with 4 µm thick and stained with hematoxylin and eosin for histological evaluation. Immunohistochemistry was performed on deparaffinized sections using an autostainer (Bond-Max autostainer; Leica Biosystems Newcastle Ltd., Melbourne, Australia) with hematoxylin as counterstain. The primary antibodies applied for confirming diagnosis were as follows: LCA (M0701, 1:200), CD1a (NCL-L-CD1a-235, 1:30), S100 (Z0311, 1:3000), and CD68 (M0814, 1:100). Appropriate positive and negative controls were utilized. The diagnoses of all patients had been confirmed by two experienced pathologists (J.-B. Liao and C.-C. Chen).

### 2.3. Mutation Analysis for KRAS, NRAS, and BRAF

Genomic DNA was extracted from formalin-fixed, paraffin-embedded blocks of all 7 cases using the QIAamp^®^ DNA FFPE Kit (Qiagen, Vialencia, CA, USA) according to manufacturer instructions. MAPK/ERK pathway mutations in either the *KRAS, NRAS* or *BRAF* gene were further screened by highly sensitive polymerase chain reaction (PCR) amplification followed by Sanger sequencing, using the *KRAS* mutant-enriched PCR Kits (FemtoPath^®^, HongJing Biotech Inc, New Taipei City, Taiwan), the *NRAS* mutation screen PCR Kits (FemtoPath^®^), and the *BRAF* mutation screen PCR Kits (FemtoPath^®^) according to manufacturer recommendations. Since sample nucleic acids with the mutation sequence were preferentially amplified over the wild type sequence, we subsequently sequenced the regions of known mutations in *KRAS*, *NRAS,* and *BRAF* using Sanger sequencing. PCR products were sequenced in the reverse direction with primers of *KRAS*, *NRAS,* and *BRAF* mentioned above and followed by Sanger sequencing (ABI3730XL, Genomics^®^,Genomics, New Taipei City, Taiwan). The sequence was interpreted by visual inspection of the DNA sequence generated by Chromas software (Technelysium Pty Ltd., Queensland, Australia).

## 3. Results

A total of 7 patients with cutaneous RDD were retrospectively recruited and analyzed with histological features, clinical characteristics, and follow-up data. The patients were four men and three women with a median age of 46 years (range: 17–62 years). All the cases suffered from persistent cutaneous lesions for at least one month, and three of them for more than one year. The cutaneous RDD was solitary in 6 patients and multifocal in 1 patient. Two patients had anemia, but one of them had a history of thalassemia. None of the patients had a fever or severe hematopoietic abnormalities. Four of seven patients (57.1%) received tumor excision without evidence of recurrence, while the other three chose oral and/or topical or intralesional corticosteroids. Case 7 received oral corticosteroids with significant clinical improvement, but case 6 received oral and topical corticosteroids with refractory treatment response. In our experience, surgery was the first-line treatment for resectable skin lesions, while topical and systemic corticosteroid therapy should be considered alternatively for unresectable cases.

A total of 9 biopsy specimens were obtained from 8 cutaneous tumors of these 7 patients. The eight tumors located on variable regions from head to lower extremities (head and neck: *n* = 3, 37.5%; lower extremities: *n* = 3, 37.5%; trunk: *n* = 2, 25%). In the skin biopsy specimens, aggregates of medium to large histiocytic cells with pale staining cytoplasm, large nuclei, and prominent nucleoli were seen in a lymphoid-rich background with occasional emperipolesis (Figure 1a,b). These histiocytic cells, the so-called RD cells, were immunohistochemically positive for S100 and negative for CD1a (Figure 1c,d).

A total of 9 specimens from 7 patients were analyzed with *BRAF, KRAS*, and *NRAS* gene mutation by highly sensitive PCR with Sanger sequencing. Mutations were detected in 5 specimens from 4 patients (4/7; 51.7%), and the genetic alterations were all detected in the *NRAS* gene. Among the 4 patients, *NRAS* A146T was the most common mutant point detected in 4 patients, followed by *NRAS* G13S detected in 2 patients. Three patients had more than one focus of genetic mutation. Table 1 summarized patients’ clinical characteristics and molecular results. Interestingly, the two biopsy specimens A and B from the same cutaneous tumor of case 6 revealed different genetic alterations, probably caused by tumor heterogeneity (Figure 2a–d).

## 4. Discussion

RDD is a rare, indolent, and self-limiting histiocytic proliferative disorder that commonly arises in young adults in the first two decades of life with a slight male predominance (M/F = 1.4:1) [5,6]. Extranodal involvement had been reported in approximately 40% of cases [7,8]. Of the extranodal involvements, the skin has been reported to be the most common site. Though extranodal involvement is not uncommon, purely extranodal RDD without lymph node involvement is rare. In 1978, Thawerani et al. firstly reported ten RDD patients with cutaneous manifestation and one of them had purely cutaneous involvement. In the ten cases, a wide range of clinical presentations were noted, including xanthomatous or erythematous rash/papules, exanthema with brownish pigmentation, pink-bluish infiltrative plaque, and nodules, etc., that can be solitary, clustered, or dispersed [10]. Later, some studied were done in purely cutaneous RDD, which were found to be distinct from typical RDD [9,13,36]. Nodules, papules, and plaques seem to be the main clinical patterns of cutaneous RDD, which may associate with purple/brown discoloration, erythema, or hyperpigmentation. Besides, it tends to affect older people (mean age: in the fourth decades) with a slight female preponderance (1.3 to 2:1). Moreover, the occurrence of associated systemic symptoms, cervical lymphadenopathy, and hematological and immunological disorders was lower [9,10,11,12,13].

Histopathological features of typical RDD are characterized by proliferation of RD cells, accompanying background infiltration of plasma cells, lymphocytes, eosinophils, or neutrophils. Emperipolesis is another classic finding in RDD, meaning the presence of inflammatory cells, such as lymphocytes or neutrophils, in the cytoplasm of RD cells [5,9,13,37]. The immunophenotype of the RD cells uniquely expresses S100, and CD68, but not CD1a in contrast to LCH. Other positive markers, including CD 163, CD 14, CD163, α1-antichymotrypsin, α1-antitrypsin, fascin, and HAM-56, had also been reported [37,38,39,40]. Compared with typical RDD, purely cutaneous RDD has similar histopathological characteristics [41]. In detail, it seems to have less emperipolesis, more frequent plasma cell infiltrates, and a greater degree of stromal fibrosis, which may result in a strikingly storiform growth pattern or a lobulated pattern [9,10,11,13,41]. Histological diagnostic criteria for cutaneous RDD were reported by Chu et al. (Table 2) [11]. A previous study published by Wang et al. demonstrated the different histological stages of purely cutaneous RDD, showing that the early or growing skin lesions presented as hypervascular architectures with mild or free of fibrosis, and relatively more plasma cell infiltrates when compared with old lesions [9].

To date, there is no standard treatment developed for RDD. This disease is thought to be self-limiting, and aggressive intervention is not required. However, some patients may present with refractory or persistent lesions. Surgery, systemic corticosteroids, cryotherapy, radiotherapy, intralesional or topical corticosteroid, interferon, dapsone, acitretin, or methotrexate had been reported for cutaneous RDD [9,42,43,44,45,46,47,48,49,50]. Surgery seems to be the exclusively effective treatment [43], though recurrence after excision had been reported [9]. Besides, surgical intervention is more suitable for those with a single or smaller lesion, and systemic treatments should be considered in cases with a larger lesion.

The pathophysiology of RDD remains unclear. Viral infection such as herpesvirus-6 and Epstein-Barr virus is supposed to be one of the etiologies resulting in host immune dysregulation [14,15,16]. Germline mutation in SLC29A3 was also reported in some familial cases with controversy [17,18]. In the current literature, investigations for gene mutations in RDD are limited and the results are still controversial. One previous study reported a patient with autoimmune lymphoproliferative syndrome developing histiocytic sarcoma in a background of RDD with a germline missense mutation in exon 9 of the TNFRSF6 gene encoding Fas, which elucidated a probable Fas/Fas ligand pathway–mediated pathogenesis [51].

The MAPK/ERK pathway, also known as the RAS-RAF-MEK-ERK pathway, regulates many fundamental cellular processes, including cell proliferation, survival, differentiation, apoptosis, motility, and metabolism. A small G protein (RAS) and three protein kinases (RAF, MEK, ERK) are the general structures involved in this pathway [52,53]. Aberrant activation of the pathway is frequently seen in human cancers, including colon cancer, pancreatic cancer, melanoma, papillary thyroid cancer, etc. With the application of recent breakthroughs in molecular pathology, genomic mutations were identified in more and more diseases. Recently, some specific mutations in the MAPK/ERK pathway have been identified in LCH, ECD, and other non-LCH diseases [20,21,22,23,24,25,26,27,28]. These results indicated that the genomic mutation in the MAPK/ERK pathway may play a role in the pathogenesis of these histiocytic disorders. Because of the high association between the MAPK/ERK pathway and histiocytosis, RDD, an entity of histiocytic proliferative disorders, was investigated. Table 3 summarizes the literature review of gene mutations described in RDD [21,28,29,30,31,32,33,34,35,51,54,55,56]. Approximately, 33–50% of RDD cases were detected with the genomic mutation involving *KRAS, NRAS, ARAF,* and *MAP2K1* [21,28,29,30,31,32,33], and only two cases with *BRAF* mutation were reported [34,35].

With the possible pathomechanism of MAPK/ERK pathway in RDD, the potential risks of malignant associations were concerned as in LCH [57] and non-LCH, such as ECD [58]. In RDD, malignant transformation to histiocytic sarcoma was reported in two cases [51,59], and RDD with concomitant malignant lymphoma was identified in 27 cases [60,61]. However, as mentioned earlier, one of the cases transforming to histiocytic sarcoma had a history of autoimmune lymphoproliferative syndrome with a germline missense mutation in the TNFRSF6 gene, making the pathogenesis of malignant transformation more complicated [51]. The mechanism of RDD coexisting with lymphoma is still unclear, which could be an aberrant histiocytic response to EBV infection. But, the most common type of lymphoma concurrently involving the same lymph node in RDD was nodular lymphocyte-predominant Hodgkin lymphoma, a histological subtype of Hodgkin lymphoma rarely associated with EBV [61]. Another conceivable mechanism is the development of RDD-like changes resulting from the secretion of certain cytokines (such as interleukin) and growth factors by lymphoma cells. However, there is some doubt as to whether it is a morphologic mimic of RDD or truly RDD [61,62,63]. Furthermore, environmental exposures may also contribute to tumorigenesis or act as the second hit to the susceptible individuals. In LCH, for example, more than 50% of cases are identified with *BRAF* mutation [20], and approximately 30% of *BRAF* wild-type LCH with *MAP2K1* mutation [64], but the prevalence of additional malignancies (32%) is lower than that of genome mutations. This result implied that genome mutation alone may not be enough to cause additional malignancies. Ma J et al. demonstrated that adult LCH patients with additional malignancy of lung cancer had a high percentage of smokers (5/6, 83%) [57], indicating the additional role of environmental exposures in tumorigenesis. Although the relationship between mutations in the MAPK/ERK pathway and potential malignant change or concomitant malignancy of RDD remains unclear, it provides a clue of the possible instability of the genome. Combining the shreds of evidence discussed above, the MAPK/ERK pathway in RDD may direct a novel and promising aspect of treatments in the future. Jacobsen et al. reported a case of RDD with *KRAS* mutation successfully treated with cobimetinib, a MEK inhibitor, which provided further support of the importance of the MAPK/ERK pathway in RDD [33].

In this study, the median age of our cases was 46 years which is compatible with the previous studies of purely cutaneous RDD. Our case study had a slight male predominance (four men and three women), and this opposite result may be related to the small case number. The clinical lesions were variable but nodules or plaques were the main presentations. A palpable lymph node was not identified in our cases, although small reactive lymphadenopathy over bilateral inguinal areas was seen in case 6. Anemia was noted in two patients, one of whom had a past history of thalassemia. Excluding the underlying disease, 4 out of 7 patients had no accompanying systemic symptoms, lymphadenopathy, and hematological and immunological abnormalities. After follow-up, none of our cases has evidence of malignant transformation.

In our cases, 4 out of 7 patients (57.1%) were detected with genetic mutations and all the mutant regions occurred in the *NRAS* gene. No mutation was identified in *KRAS* and *BRAF*. All 4 cases with *NRAS* mutations had point mutation at *NRAS* A146T. *NRAS* G13S mutation was found in 2 patients, while *NRAS* G12D, *NRAS* G12N, and *NRAS* A59V were observed once in different specimens. Interestingly, different genetic mutations were found in case 6, who had two specimens (specimen A with *NRAS* G12D, and specimen B with *NRAS* G12N and *NRAS* A146T). Referring to her clinical presentation, the two specimens were taken from different components of the cutaneous tumor. The different gene mutations in the same patient may be related to the heterogeneity of tumor cells. There was no significant difference in clinical presentation in cases with or without *NRAS* mutation. Comparing with previous studies, we could find that *NRAS* mutation was not frequently identified in patients with RDD. Only Diamond et al. [28] reported one patient with *NRAS* mutation, but there was no further information on the involved area, genomic coordinates, and variants. Furthermore, most of the mutant reported cases were not presented with cutaneous lesions. Therefore, the role of *NRAS* mutation in RDD, or specifically cutaneous RDD remains unknown. The *NR**AS* protein encoded by the *NRAS* gene is a GTPase, acting as a switch for pathway activation and playing a crucial role in cell proliferation and survival. When it gets mutated, it can act as an oncogene, which can be seen in some malignancies, such as melanomas and some hematopoietic malignancies [65]. Congenital melanocytic nevi were also found to frequently harbor *NRAS* mutations [66]. Some studies mentioned that *NRAS* mutation seemed to be a poor prognosis in leptomeningeal melanosis and melanomas because of the activation of both ERK and the phosphatidylinositol 3-kinase (PI3K)/AKT pathway, in contrast to ERK signaling alone in *BRAF* mutation cases [67,68]. Whereas, in metastatic colorectal cancer, *BRAF* mutation seems to have the poorest prognosis compared with *KRAS* and *NRAS* [69]. Of note, most of the reported *NRAS* mutations are located at codons 12, 13, and/or 61, not NRAS A146T [70]. Besides, Garces et al. found that in RDD, immunostain for phospho-ERK overexpression was seen in all *MAP2K1*-mutated cases of RDD, supporting the activation of MAPK/ERK pathway, but not in *KRAS* mutated cases. These results may indicate that there may be some differences in the downstream cascade activation or even an alternative pathway in *KRAS*-mutated RDD [30]. Is there also an alternative pathway in *NRAS*-mutated RDD? Does this alternative mechanism make any difference in the clinical presentation and prognosis or let tumor cells be more confined to the skin? These questions remain unclear.

Compared with other mutations in MAPK/ERK pathway, *NRAS* mutation is less frequent [69,71], either in histiocytic disorders or other malignancies. However, the percentage of *NRAS* mutation is considerably high (57.1%) in our study. All the cases in our study were Taiwanese people. Can *NRAS* mutation be an ethnic mutation or a common single nucleotide polymorphism in Asians? Unfortunately, there is no population-based genomic study of *NRAS* mutation in the general Asian population or other ethnicities so far. However, in malignancies with frequent mutations in the MAPK/ERK pathway, such as melanomas and colon cancers, the percentage of *NRAS* and *KRAS* mutation in Asians is not higher than those in Caucasians. In fact, *NRAS* mutation is even higher in Caucasian patients with melanomas. In addition, our case number is too small to demonstrate the possible ethnic effect. Besides, the highly sensitive PCR were all performed by a Taiwanese company, so machine bias cannot be totally excluded, but which is less likely because there is no *NRAS* mutation detected in 3 cases [72,73,74,75]. Owing to the limited data, further studies and more cases are needed for the impact of *NRAS* mutation in tumorigenesis and prognosis.

To our knowledge, this is the first study to demonstrate gene mutations in cutaneous RDD. *NRAS* mutation was frequent in cutaneous RDD, especially at *NRAS* A146T. Our results indicate that *NRAS* mutation in MAPK/ERK pathway may be involved in the pathogenesis of RDD, especially in cutaneous RDD. To sum up our findings, MAPK/ERK pathway seems to be an important pathogenic pathway in a part of the patients with RDD. Our hypothetical thesis of molecular pathogenesis for cutaneous RDD is shown in Figure 3.

There are several limitations in the present study. First, this study is performed retrospectively and the DNA quality of paraffin blocks might not be as good as fresh tissues. Second, the case number is small, and the findings needs to be confirmed in large case series. Third, all the cases in this study are Asians. Whether these results can be applied in other populations such as Caucasian remains uncertain. Future studies are needed. Fourth, the follow-up period is short that most of the patients lost follow-up several months after the diagnosis.

## 5. Conclusions

In conclusion, our study supports that cutaneous RDD is a distinct group apart from typical RDD. Cutaneous RDD tends to affect older people, and the histopathological features showed less emperipolesis, more dominant plasma cell infiltrate, and relatively more stromal fibrosis. The etiology of RDD remains unclear, but our molecular results provide an implication that MAPK/ERK pathway may play an important role in the pathogenesis of cutaneous RDD, especially *NRAS* mutation. *NRAS* A146T was the most frequent genomic mutation in our cases. Patients with refractory cutaneous RDD having mutation of MAPK/ERK pathway may benefit from targeted treatment. Further large studies with a greater sample size of patients are required to ascertain the etiology and pathogenesis of RDD or cutaneous RDD.

## Figures and Tables

**Figure 1 biology-10-00396-f001:**
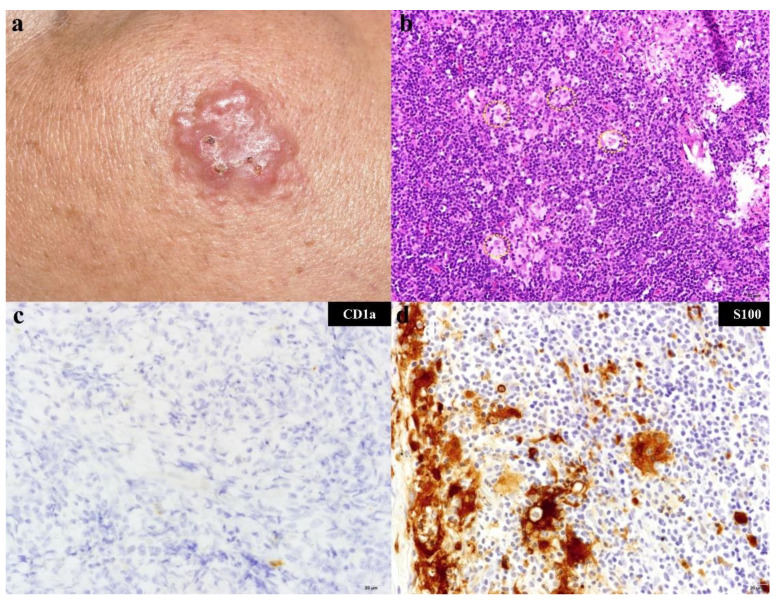
Clinical and pathologic features of the case 4. (**a**) The patient was a 46-year-old man who presented with a mild itchy erythematous nodular plaque on the right cheek for 5 months; (**b**) In the skin biopsy specimen, aggregates of Rosai–Dorfman (RD) cells with pale staining cytoplasm, large nuclei, and prominent nucleoli were seen in a lymphoid-rich background. Emperipolesis was noted (yellow circle, 200×). Scale bar = 50 μM; (**c**) Immunohistochemistry of CD1a was negative (400×). Scale bar = 20 μM; (**d**) S100 staining was positive in RD cells (400×). Scale bar = 20 μM.

**Figure 2 biology-10-00396-f002:**
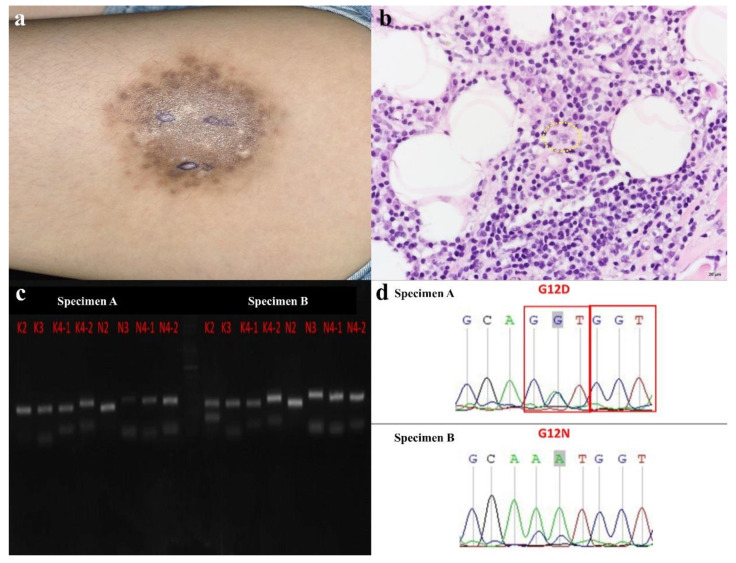
Clinicopathologic features and molecular analysis of the case 6. (**a**) The patient was a 17-year-old girl who presented with one 8 × 8 cm tender hard brownish plaque with some whitish papules and nodules on her left thigh for 3 years. We took two different specimens, specimen A from black parts and specimen B from white parts, for further evaluation. (**b**) In the skin biopsy, RD cells with emperipolesis were found in a prominent lymphoplasmacytic background (yellow circle, 400×). Scale bar = 20 μM. (**c**) In both specimens A and B, the agarose gel electrophoresis of PCR products for screening *KRAS* and *NRAS* mutation were present. (**d**) Further sequencing revealed *NRAS* G12D mutation in specimen A, and *NRAS* G12N mutation in specimen B.

**Figure 3 biology-10-00396-f003:**
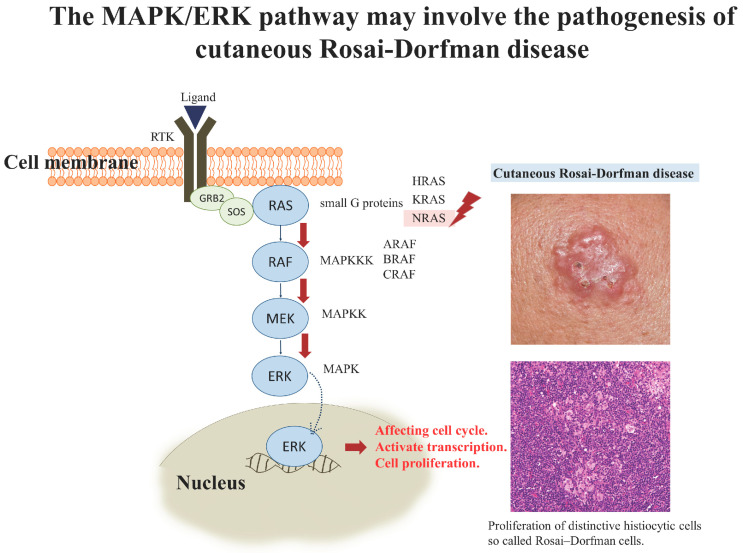
Our hypothetical thesis of molecular pathogenesis of cutaneous RDD. In MAPK/ERK pathway, a signaling cascade is initiated when ligands bind to two receptor tyrosine kinases (RTK) subunits and induce receptor dimerization. Adaptor proteins such as growth factor receptor binding protein 2 (GRB2) result in the recruitment of guanine nucleotide exchange factors (GEFs) such as Son of Sevenless (SOS) protein to the cell membrane. SOS then activates RAS by facilitating the small G proteins, including RAS subfamily GTPases such as HRAS, KRAS, and NRAS, to dissociate from GDP and bind to GTP. GPT-bound RAS then activates RAF, a subfamily of MAPKKK, which phosphorylates MEK (MAPKK). Phosphorylated MEK in turn activates ERK (MAPK). Activated ERK enters the nucleus, where it phosphorylates numerous nuclear transcription factors and modulates gene expression, which regulates many cellular processes, including cell proliferation, survival, differentiation, apoptosis, motility, and metabolism. When *NRAS* gene gets mutated, the signal pathway will lose its normal regulation, which activates the downstream cascade resulting in the proliferation of distinctive histiocytic cells in cutaneous RDD.

**Table 1 biology-10-00396-t001:** Summary of patients’ clinical characteristics and molecular data.

Patient No	Age/Gender	Site	Presentation	Associated Finding	Treatment	Gene Mutation
1	47/Female	Right Leg	One Painful Flesh-Colored Nodule	Anemia (Hb: 8.6, Mcv: 64.8), History of Thalassemia	Excision	No Mutation was Detected
2	48/Male	Left Back and Left Neck	One 12 × 5 cm Erythematous Plaque on Left Neck and one 6 × 3 cm Erythematous Plaque on Left Back for one More Year	No Systemic Symptom	Excision	No Mutation was Detected
3	62/Male	Left Back	One Soft and Tender Subcutaneous Nodule for 2 Years	No Systemic Symptom	Excision	No Mutation was Detected
4	46/Male	Right Cheek	One Mildly Itchy Erythematous Nodular Plaque for 5 Months	No Systemic Symptom	Biopsy and Intralesional Corticosteroid Injection	*Nras* G13s, *Nras* A59v, *Nras* A146t
5	32/Female	Right Zygomatic Area	One 0.6 cm Subcutaneous Nodule	Steatocystoma Multiplex Polychondritis	Excision	*Nras* G13s, *Nras* A146t
6	17/Female	Left Thigh	One 8 × 8 cm Tender Hard Brownish Plaque with Some Whitish Component for 3 Years	Small but Palpable Reactive Lymphadenopathy on Bilateral Inguinal Areas and Then Remission	Oral and Topical Corticosteroids	Specimen A: *Nras* G12d; Specimen B: *Nras* G12n, *Nras* A146t
7	38/Male	Left Thigh	One 7 × 7 cm Flesh-Colored Asymptomatic Indurated Subcutaneous Mass for 1.5 Months	Anemia (Hb: 12.8, Mcv: 87.1)	Oral Corticosteroids	*Nras* A146t

**Table 2 biology-10-00396-t002:** **A** histologic summary of cutaneous Rosai–Dorfman disease (modified from reference [11]).

**Major Characteristics**
Infiltrates or aggregates of atypical histiocytic cells with abundant cytoplasm with irregular borders, round vesicular nuclei, and small nucleoli. Emperipolesis (intact cells within the cytoplasm and surrounded by small halos) of atypical histiocytic cells may be seen.
2. Dense infiltrates of lymphocytes and plasma cells, admixed with neutrophils and eosinophils, and stromal fibrosis.
3. Prominent venules infiltrated by plasma cells, more prominent at peripheral areas.
4. Lymphoid aggregates and germinal center formation, more prominent at peripheral areas.
5. Lymphatic involvement of atypical histiocytic cells, more frequent at central areas.
**Minor Characteristics**
Pseudoepitheliomatous hyperplasia.
2. Exophytic growth with thinning of overlying epidermis.
3. Eosinophilia.

**Table 3 biology-10-00396-t003:** The summary of genetic mutations described in RDD.

Reference	Case Number of RDD	Primary Site of RDD	Result of Gene Mutation	Ref No
Venkataraman et al., 2010	1	Axillary Lymph Node ^1^	Germline Missense Mutation in Exon 9 of the TNFRSF6 Gene.	[51]
Haroche et al., 2012	23	Not Available.	*BRAF*: 0%	[21]
Diamond et al., 2016	8	Lymph Node: 4 cases; Pituitary Stalk: 1 case; Cheek: 1 case; Meninges: 1 case; Axillary soft tissue: 1 case.	*KRAS*: 2 cases (25%); *NRAS*: 1 case (12.5%); *ARAF*: 1 case (12.5%); Wild Type: 4 cases (50%).	[28]
Shanmugam et al., 2016	1	Submandibular Salivary Gland	*KRAS* K117N	[29]
Garces et al., 2017	21	Lymph Node: 8 cases; Soft Tissue: 7 cases; Breast: 3 cases; Bone: 3 cases; Nasopharynx: 1 case.	Genetic Mutation was Detected in 7 cases (33%) ^2^, including *KRAS* (*n* = 4) and *MAP2K1* (*n* = 3). No mutation was Identified in *ARAF*, *BRAF, PIK3CA*, or any Other Genes Assessed.	[30]
Matter et al., 2017	1	Buttock Subcutaneous Tumor with Multiple Hypermetabolic Lesions of the Bones and Bone Destruction.	*MAP2K1* L115V Mutation	[31]
Lee et al., 2017	11	Not available.	Mutation was Detected in 5 cases: 1) case 1: *KRAS* K117N; 2) case 2: *CBL* C384Y, *GNAQ* Q209H, *KRAS* K117N, *KRAS* A146V; 3) case 3: *KRAS* K117N; 4) case 4: *KDM5A* amplification; 5) case 5: *FBXW7* E113D. No Mutation was Detected in Other 6 cases.	[32]
Jacobsen et al., 2017	1	Perirenal Soft Tissue Mass	*KRAS* codon 12 (p.G12R)	[33]
Fatobene et al., 2018	1	Cervical Lymph Node	*BRAF* V600E Mutation	[34]
Choi et al., 2018	6	All Are Not Purely Cutaneous RDD	*BRAF*: 0%	[54]
Richardson et al., 2018	1	Central Nervous System	Deletion in the β3-αC Loop of the Kinase Domain in exon 12 of *BRAF*	[35]
Tanaka et al., 2019	1	Kidney	*K-* and *N-RAS*: 0%	[55]
Janku et al., 2019	3	Not mention	One RDD Patient Harbored a *CAPZA2-BRAF* Fusion and a *RAF1* Amplification; the Other 2 RDD patients had a *GNAS* R201C and *APC* E1157fs Mutation, respectively.	[56]

^1^ This patient had autoimmune lymphoproliferative syndrome and developed histiocytic sarcoma in a background of RDD. ^2^ One of the 7 mutant cases presented with a soft tissue lesion.

## Data Availability

The data presented in this study are available on request from the corresponding author. The data are not publicly available due to privacy.

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
