# Peer review of "NRAS Mutations May Be Involved in the Pathogenesis of Cutaneous Rosai Dorfman Disease: A Pilot Study"

_biology, 2021, doi:10.3390/biology10050396_

Round 1

Reviewer 1 Report

Thanks fort he opportunity to review this very interesting article on a rare disease, with only cutaneous lesions which is a quite controversial entity. The case are well documented and the results valuable. However, the reading is somewhat difficult for several grammar and style errors, which needs a careful revision. Also some repetitions can be avoided to make the reading more affordable. 

Author Response

Point 1:Thanks fort the opportunity to review this very interesting article on a rare disease, with only cutaneous lesions which is a quite controversial entity. The case are well documented and the results valuable. However, the reading is somewhat difficult for several grammar and style errors, which needs a careful revision. Also some repetitions can be avoided to make the reading more affordable. 

  • Thank you for your kind words and professional recommendation. Based on your suggestion, we had corrected the grammar mistakes by the software, Grammarly, to detect spelling, punctuation, and other common errors in texts. And a native English speaker was subsequently requested to help us proofread the manuscript. All revised parts were highlighted in red color in the revised manuscript. Please see the attachment. We’re grateful for your sincere comments.

Reviewer 2 Report

In the study entitled: "NRAS mutations may be involved in the pathogenesis of cutaneous Rossi Dorfman disease" the authors present a communication paper on the very rare Rossi-Dorman disease (RDD). They limit themselves to purely cutaneous RDD and examine tissue samples from a total of seven patients for gene mutations using PCR analysis and Sanger sequencing. Due to the small number of cases (n=7), the work should be considered as a pilot study, which should also be reflected in the title. In the introduction, the authors go well into the disease and the problems associated with it using the relevant literature. The methodological structure is clear and comprehensibly described. In the results section, the authors' own results are well described and reconciled with the current literature in the discussion section. The authors also address the limitations of the work. Overall, this is an easily readable article about a very rare disease that may provide further clues about the pathomechanism, which, however, should be validated in further studies on a larger patient population. I recommend the work for publication after minor modifications.

Author Response

Point 1: In the study entitled: "NRAS mutations may be involved in the pathogenesis of cutaneous Rossi Dorfman disease" the authors present a communication paper on the very rare Rossi-Dorman disease (RDD). They limit themselves to purely cutaneous RDD and examine tissue samples from a total of seven patients for gene mutations using PCR analysis and Sanger sequencing. Due to the small number of cases (n=7), the work should be considered as a pilot study, which should also be reflected in the title.

  • Thank you for your professional recommendation. Indeed, since the purely cutaneous RDD is rare, the case number is small. That’s why we submitted the manuscript as a Communication article to be a short brief or pilot study. Based on your suggestion, we revised the title with a special mention of “a pilot study” highlighted in red color in our revised manuscript. Please see the attachment.

Point 2. In the introduction, the authors go well into the disease and the problems associated with it using the relevant literature. The methodological structure is clear and comprehensibly described. In the results section, the authors' own results are well described and reconciled with the current literature in the discussion section. The authors also address the limitations of the work. Overall, this is an easily readable article about a very rare disease that may provide further clues about the pathomechanism, which, however, should be validated in further studies on a larger patient population. I recommend the work for publication after minor modifications.

  • Thank you for your kind words and sincere comments. Definitely, our findings need large case series of RDD for further testing the genetic alterations in the MAPK/ERK pathway. Our study should be taken as a pilot study to turn on the lights of discussion on the possible oncogenetics. Please see the attachment.

Reviewer 3 Report

1. The manuscript is very well written.

2. What is "NRAS"? You have it on your title, in the text, but nowhere in full text (from what I can see). Since it is in the foreground, the meaning should be described in parentheses at least in the abstract.

3.  NRAS may be an ethnic mutation? It is a serious question that deserves a hypothesis in the main text (since all patients are white asian). I am asking because of the papers from below.

4. On the conclusion subsection: "... our molecular results provide an implication that MAPK/ERK pathway may play an important role in the pathogenesis of cutaneous RDD". The same conclusions are presented in these papers also:

https://www.ncbi.nlm.nih.gov/pmc/articles/PMC5837474/

https://www.nature.com/articles/modpathol201755

https://onlinelibrary.wiley.com/doi/pdf/10.1111/bjh.16006

https://www.x-mol.com/paper/853802

Please put "NRAS" first in the phrase and only after mention the MAPK/ERK pathway. "NARS" is important here.

5. Usually 7 cases would be few, that's what anyone would say, but even so, they are still useful.

Overall, I consider that this manuscript can be published as is. It definitely adds value to the subject in question.

Author Response

Point 1. The manuscript is very well written.

  • Thank you for your kind words. We’re sincerely grateful.

Point 2. What is "NRAS"? You have it on your title, in the text, but nowhere in full text (from what I can see). Since it is in the foreground, the meaning should be described in parentheses at least in the abstract.

  • Thank you for your professional recommendation. The NRAS proto-oncogene was named by its encoding protein, neuroblastoma RAS viral oncogene homolog, which is a membrane protein and has intrinsic GTPase activity. NRAS protein can be activated by a guanine nucleotide-exchange factor and inactivated by a GTPase activating protein. Mutations in the NRAS gene can activate the RAS/MAPK signaling and have been reported to be associated with various cancers, e.g. rectal cancer and follicular thyroid cancer. Based on your professional suggestion, we openly spelled the genes and briefly introduced the importance in the abstract of our revised manuscript. All revised parts were highlighted in red color. Please see the attachment.

Point 3.  NRAS may be an ethnic mutation? It is a serious question that deserves a hypothesis in the main text (since all patients are white asian). I am asking because of the papers from below.

  • Thank you for your professional recommendation. Is the NRAS mutation an ethnic mutation? Honestly, we didn’t know, since our data only reflects the results of 7 cases with cutaneous Rosai-Dorfman disease. To date, there is no population-based genomic study on NRAS mutation in the general Asian population or other ethnicities so far. However, in malignancies with frequent mutations in the MAPK/ERK pathway, such as melanomas and colon cancers, the percentage of NRAS and KRAS mutation in Asians is not higher than those in Caucasians. In fact, NRAS mutation is even higher in Caucasian patients with melanomas. Also, our limited cases cannot make any robust conclusion, either in the pathogenesis or in the ethnic issue. We had addressed our limitation in the discussion and added one paragraph in red color to point out your concerns. Besides, we are sorry that we didn’t see any links of papers listed in this question. If you meant the papers listed in the next question, none of them have mentioned the NRAS genetic characteristics in Asians. We sincerely thank you for your scientific comments. Please see the attachment.

Point 4. On the conclusion subsection: "... our molecular results provide an implication that MAPK/ERK pathway may play an important role in the pathogenesis of cutaneous RDD". The same conclusions are presented in these papers also:

https://www.ncbi.nlm.nih.gov/pmc/articles/PMC5837474/

https://www.nature.com/articles/modpathol201755

https://onlinelibrary.wiley.com/doi/pdf/10.1111/bjh.16006

https://www.x-mol.com/paper/853802

Please put "NRAS" first in the phrase and only after mention the MAPK/ERK pathway. "NARS" is important here.

  • Thank you. Most of the articles you listed had been cited in our references, and we have similar conclusions on the genetic alterations in the Rosai-Dorfman disease. Based on your suggestion, we had open spelled the NRAS" first in the phrase of the introduction and after mention the MAPK/ERK pathway. The revised parts were highlighted in red color. Please see the attachment.

Point 5. Usually 7 cases would be few, that's what anyone would say, but even so, they are still useful. Overall, I consider that this manuscript can be published as is. It definitely adds value to the subject in question.

  • Thank you for your kind words and professional comments.
